# Highly Sensitive Temperature Sensors Resulting from the Luminescent Behavior of Sm^3+^-Doped Ba_2_MgMoO_6_ High-Symmetry Double-Perovskite Molybdate Phosphors

**DOI:** 10.3390/ma17081897

**Published:** 2024-04-19

**Authors:** Natalia Miniajluk-Gaweł, Bartosz Bondzior, Maciej Ptak, Przemysław Jacek Dereń

**Affiliations:** Institute of Low Temperature and Structure Research, Polish Academy of Science, Okolna 2, 50-422 Wroclaw, Poland; b.bondzior@intibs.pl (B.B.); m.ptak@intibs.pl (M.P.)

**Keywords:** double-perovskite, molybdate phosphors, high symmetry, WLED, optical thermometer

## Abstract

We present double-perovskite molybdate with the formula of Ba_2_MgMoO_6_ doped with Sm^3+^ ions as a potential red phosphor to improve the color characteristics of white-light-emitting dioded (wLEDs). The new orange–red phosphor was synthesized using the co-precipitation (CP) method, and then its structural and spectroscopic properties were determined. Red emission at 642.6 nm dominates, which results from the electric dipole (ED) transition of the ^4^G_5/2_ → ^6^H_9/2_ type, and the materials are characterized by short luminescence decay times. BMM:Sm^3+^ is, to our best knowledge, the clearest example of dominant red emission of Sm^3+^ resulting from the location of the dopant in octahedral sites of high-symmetry cubic structure. In the sample containing 0.1% Sm^3+^, Sm^3+^ ions are located in both Mg^2+^ and Ba^2+^ sites, while at higher concentrations the Ba^2+^ site is less preferable for doping, as a result of which the emission becomes more uniform and single-site. The relative sensitivity calculated from FIR has a maximum of 2.7% K^−1^ at −30 °C and another local maximum of 1.6% K^−1^ at 75 °C. Such value is, to the best of our knowledge, one of the highest achieved for luminescent thermometry performed using only Sm^3+^ ions. To sum up, the obtained materials are good candidates as red phosphor to improve the color characteristics of wLEDs, obtaining a color-rendering index (CRI) of 91 and coordinated color temperature (CCT) of 2943 K, constituting a warm white emission. In addition to this, a promising precedent for temperature sensing using high-symmetry perovskite materials is the high sensitivity achieved, which results from the high symmetry of the BMM host.

## 1. Introduction

Molybdates are materials based on an oxyanion group Mo_x_O_y_, finding applications in fields like paramagnetic materials [1], catalysts [2], and optical materials [3,4]. Recently, the interest of the research community has grown as the double-perovskite materials were seen as promising due to their highly symmetrical structure, and molybdates are one of the most popular types of inorganic materials found to exhibit the double-perovskite structure described with the formula A_2_BB’O_6_, where A sites are occupied by alkaline metals or lanthanides, while positions B and B’ are reserved for transition metal ions. This arrangement creates a more complex and layered crystal lattice compared to the simple perovskite structure. Double-perovskite molybdates exhibit high thermal and chemical stability, and suitability for the luminescent dopants [5], which set that class of materials apart from other double-perovskites, such as halides.

Examples of these materials include Sr_2_FeMoO_6_, known for its colossal magnetoresistance (CMR) effect, which makes it a promising candidate for applications in spintronics and magnetic sensors [6], and Ba_2_PrMoO_6_, explored for its photocatalytic and photocurrent properties, making it relevant for applications in solar energy conversion and water splitting [7].

The research on the luminescence properties of double-perovskite molybdates is limited, with most of the published articles being focused on Eu^3+^ luminescence [5,8,9,10]. It is apparent that the rare earth dopants tend to locate themselves in highly symmetrical octahedral MgO_6_ sites, which are credited for their unique luminescence properties in the form of almost-monochromatic emission spectra [8,9]. This work concerns the Ba_2_MgMoO_6_ double-perovskite, and so far, only three publications [9,11,12] have been published on Eu^3+^ doping. It should be emphasized that this is the first work that describes the physicochemical properties of this matrix doped with Sm^3+^ ions.

Mo-based double-perovskite phosphors are of great interest due to reliable, lasting and stable emission, easy fabrication, low cost, high durability, and strong absorption in the NUV region [11,12,13]. The Sm^3+^ ion was used as an activator of phosphors because these ions have a rich structure of energy levels and can transfer electrons through the 4f-4f transitions. This makes Sm^3+^-doped phosphors generally exhibit multiple absorption peaks in the range of 220–550 nm, and thus they can be excited effectively using a near-UV or blue LED chip. These phosphors emit red–orange light in the visible range with four narrow emission bands, thanks to the characteristic transitions ^4^G_5/2_ → ^6^H_J_ (J = 5/2, 7/2, 9/2 and 11/2) [14,15].

Temperature is one of the most important physical parameters, so its precise measurement is extremely important. Non-contact luminescent thermometers, compared to traditional contact thermometers, offer many advantages such as non-invasive measurement, fast response, high sensitivity, ability to be used in rapidly changing conditions, and strong electromagnetic fields. The most commonly used temperature reading is based on the relative intensity ratio of two different emission bands, because this method minimizes the influence of disturbing external factors [9,16]. It turned out that the material we proposed, apart from being an excellent material for improving the quality of WLED lighting, can also be successfully used as a highly sensitive optical thermometer, which results directly from the high symmetry. The current state of the research on molybdate phosphors for temperature sensors is not extensive and mainly concerns alkaline earth molybdates [17,18,19]. However, there is no information about double-perovskites, so this work is very valuable and provides new information regarding non-contact thermometry.

In this paper, double-perovskite molybdate with the formula Ba_2_MgMoO_6_ is studied as a host for Sm^3+^ and as a potential red phosphor to improve the color characteristics of wLEDs. For Sm^3+^-doped materials for wLED, the main challenge is to obtain mostly red emission at ~645 nm at the expense of the yellow (570 nm), orange (600 nm), and NIR (710 nm) emissions [20]. We synthesized a novel orange–red phosphor using the co-precipitation method, and then investigated the luminescence properties of these phosphors as representative materials. Ba_2_MgMoO_6_ double-perovskite as a host for Sm^3+^ ions has the advantage of high site symmetry and host-sensitization in the near-UV region. Additionally, the obtained temperature measurements results showed that Ba_2_MgMoO_6_: Sm^3+^ double-perovskite has a great potential for applications in non-contact optical thermometry.

### 1.1. Experimental

#### Samples Preparation

The co-precipitation synthesis method was used to synthesize a series of Sm^3+^-doped Ba_2_MgMoO_6_ (BMM) double-perovskite samples, described using the formula Ba_2_Mg_1−x_Sm_x_MoO_6_ (x = 0.1–1%).

Ba(CH_3_COO)_2_ (Alfa Aesar (Ward Hill, MA, USA), 99%), Mg(CH_3_COO)_2_·4H_2_O (Alfa Aesar, 99.95%), (NH_4_)_6_Mo_7_O_24_·4H_2_O (AMT) (Sigma–Aldrich (St. Louis, MO, USA), 99.99%), Sm(NO_3_)_3_ (Alfa Aesar, 99.9%) were used as starting materials in the co-precipitation method. Li_2_CO_3_ (Alfa Aesar, 99.998%) was applied to charge compensation in a ratio of 1:1, based on the amount of dopant ions. To compensate for the evaporation of magnesium ions during the sintering process, a 20% excess of magnesium ions was applied. Firstly, the stoichiometric amounts of precursors were dissolved separately in distilled water. Next, a white precipitate was formed immediately after the first droplet of AMT solution was added slowly (2 mL/min) under stirring at 200 rpm, 25 °C to the remaining clear solution. The water was evaporated and the precipitate was dried by heating at 80 °C for 20 h and then pre-sintering at 600 °C for 12 h. The final annealing was carried out at 1150 °C for 6 h with a constant heating rate of 3 °C/min using corundum crucible. After each step, the obtained products were ground for 15 min.

### 1.2. Research Techniques

X-ray diffraction (XRD) patterns were recorded with X’Pert ProPANalytical X-ray diffractometer (Malvern, UK), working in the reflection geometry, using CuKα radiation (λ = 1.54056 Ǻ). The data were collected in a 2Θ range from 10° to 90° with a step of 0.026°.

Raman spectra were measured using a Renishaw inVia spectrometer (Middlewich, UK) equipped with a confocal optical microscope and a diode laser operating at 830 nm.

A scanning electron microscope (FEI NOVA NanoSEM 230 (Eindhoven, The Netherlands), equipped with EDAX Genesis XM4 detector (Eindhoven, The Netherlands)) was used to characterize the morphology and chemical composition of the samples. The SEM images were recorded with an accelerating voltage of 5 kV.

Emission spectra, excitation spectra, and decay curves were measured using an FLS1000 Edinburgh Instruments (Edinburgh, Scotland) spectrophotometer in a Czerny–Turner configuration with VIS and NIR at room temperature.

The diffuse reflection spectrum was recorded using a Varian Cary 5E UV/VIS-NIR spectrophotometer (Varian Incorporation, Palo Alto, CA, USA).

The emission spectrum as a function of temperature was measured using the Linkam THMS600 (Salfords, UK) stage, which has both heating and cooling functions and Hamamatsu R928 photomultiplier (Hamamatsu, Japan). The measurement was carried out in the range from −193 °C to 150 °C with a constant step of 20 °C.

## 2. Results and Discussion

### 2.1. Structural Properties

Figure 1a shows X-ray diffractograms of molybdate materials synthesized using the co-precipitation method, along with the visualization of the structure. There is no standard pattern for the structure of BMM, but according to what was written in the previous publication [9], the structure of this material matches well with the ICDD 070-2023 pattern of Ba_2_MgWO_6_ (see Figure 1a). It was possible to obtain practically single-phase materials with a small amount of an additional phase of BaMoO_4_ ICDD 43-0646 (less than 1%), the presence of which does not in any way affect the physicochemical properties of the obtained materials. It can be seen that, as the Sm^3+^ concentration increases, the diffractograms shift towards lower theta angles (Figure 1b). This means that, as the amount of Sm^3+^ increases, the lattice constants increase because the smaller ion in the host (Mg^2+^, IR = 86 pm) is replaced by a larger one (Sm^3+^, IR = 109 pm). The obtained materials have a cubic structure, crystallizing in the *Fm*$\bar{3}$*m* space group (Figure 1c), and are characterized by the following cell parameters: a = 8.1120 Å, unit cell volume V = 533.81 Å^3^ and Z = 4.

In the SEM image (Figure 1d), one can notice that the obtained material consists of crystallites of undefined shapes: oval with rounded edges. We determined the average crystallite sizes in the ImageJ 1.51j8 program, and they are 0.62 µm. The Appendix A contains a histogram of the average crystal size, determined on the basis of 100 crystallites (Appendix A).

To further check the quality of the samples, Raman spectra were collected (see Figure 1e). The spectrum presents two narrow bands near 126 (F_2g_) and 438 cm^−1^ (F_2g_) as well as broader Raman peaks near 780 cm^−1^ (A_1g_), which is in good agreement with the previously reported experimental and theoretical data for molybdates and tungstates having double-perovskite architecture [21,22]. The cubic *Fm*$\bar{3}$*m* structure exhibits only four optical phonons (A_1g_ + E_g_ + 2F_2g_), but the Raman band corresponding to the E_g_ mode is usually too weak to be detected [22,23].

### 2.2. Spectroscopic Properties

The absorption spectra of BMM:Sm^3+^ exhibit absorption peaks in the near infrared, where the lower-lying levels of Sm^3+^ are located (Figure 2a inset). These absorption lines are primarily due to ^6^H_5/2_ → ^6^F_J_ (J = 1/2, …, 11/2) transitions.

The emission spectra of the BMM:Sm^3+^ (Figure 2b) exhibit four emission peaks resulting from the ^4^G_5/2_ → ^6^H_J_ (J = 5/2, …, 11/2) transitions, characteristic of Sm^3+^ along with smaller neighboring peaks assigned to the vibronic transitions of the mentioned main emission lines. The optimal concentration of Sm^3+^ is 0.3% (Figure 2c).

Although the XRD results (Figure 1b) suggest that the Sm^3+^ substitutes predominantly Mg^2+^ in the structure, one can identify the two-site emission of Sm^3+^ in the emission spectra of the sample with 0.1% Sm^3+^ by comparing the relative intensity of the maxima at 597 nm and 642.6 nm; the sample with 0.1% Sm^3+^ visibly exhibit stronger 597 nm emission as compared to other Sm^3+^ concentrations. This suggests that, in this sample, the Sm^3+^ ions are located in both Mg^2+^ and Ba^2+^ sites. For higher concentrations, as the lattice constant increases the Ba^2+^ site becomes less preferable for Sm^3+^ dopant, and as a result the emission becomes more uniform and seemingly one-site. The radio between the orange and red emission is known to indicate the symmetry of the Sm^3+^ ions site; however, it is usually inferred that the red emission at 642.6 nm resulting from the electric–dipole (ED) type ^4^G_5/2_ → ^6^H_9/2_ transition is dominant at low-symmetry sites [24], and orange emission at ~600 nm resulting from the mixed electric–dipole/magnetic–dipole (ED + MD) type ^4^G_5/2_ → ^6^H_7/2_ transition is dominant at high-symmetry octahedral sites [25]. This straightforward causation is, however, contested by the results of some highly-symmetrical double-perovskite materials such as Ba_2_ZnWO_6_:Sm^3+^, which exhibits dominant red emission [26] similar to BMM:Sm^3+^, as well as comparative studies between orthorhombic LiCaBO_3_:Sm^3+^ and cubic CsCaBO_3_:Sm^3+^ [27]. In the latter, the Sm^3+^ ions exhibit dominant red emission when located in octahedral sites of CsCaBO_3_ [28] and dominant orange emission when located in 7-coordinated sites in LiCaBO_3_ [29]. BMM:Sm^3+^ is to our best knowledge the most clear example of dominant red emission of Sm^3+^ resulting from the location of the dopant in octahedral sites of high-symmetry cubic structures.

The excitation spectrum of BMM:Sm^3+^ monitored at the maximum intensity at 642 nm (Figure 3a) consists of broad bands in the UV/blue spectral region 300–450 nm, which originates from the absorption of the O^2−^ → Mo^6+^ charge transfer. The excitation spectrum monitored at the maximum of the emission of the Sm^3+^ at the second (Ba) site consists of an additional band with maximum at 260 nm, which is due to excitation into Ba states in the conduction band [9]. In the visible part of the excitation spectrum, there are weaker lines associated with the *4f-4f* transitions labeled in the graph. Based on the complimentary data from the excitation, absorption, and emission spectra, the positions of the barycenter of Sm^3+^ energy levels were determined and listed in Table 1.

The decay curves have two components: a short one and a long one (Figure 3b,c), which confirm the two-site emission of Sm^3+^. The short component has a decay time around 350 μs (Figure 3b). Such short decay times for Sm^3+^ luminescence have been observed in materials where Sm^3+^ is located at octahedral sites: CaNb_2_O_6_ [30] and TiO_2_ [31]. The decay times become increasingly shorter when the dopant concentration increases, due to concentration quenching [32]. The long component is, as expected, more dominant in decay curves monitored at 597 nm (Figure 3c), and exhibit decay time 2.6 ± 0.2 ms, observed commonly for Sm^3+^-doped materials with a site coordination number other than 6, such as SrAl_4_O_7_ [33] and Sr_2_SiO_4_ [34].

The emission spectra of BMM:Sm^3+^ measured in the −193–150 °C range (Figure 4a) reveal that both the intensity and shape of the Sm^3+^ emission is strongly temperature-dependent. The thermal quenching of the emission intensity (Figure 4b) has two components when fitted with an Arrhenius equation (Appendix A) with corresponding activation energies of 417 cm^−1^ and 2054 cm^−1^. The former component with smaller activation energy is absent, when the sample is excited at 266 nm into the Ba states (which constitute the conduction band); therefore, it is concluded that it must be associated with the quenching of the molybdate group charge transfer state. This is depicted in Figure 4c as a crossing between the ground state and the excided state parabolas. Similarly, low activation energy of thermal quenching was observed for the pure BMM host and Eu^3+^–doped BMM [9]. The latter and larger activation energy can be matched with the thermalization of the ^4^G_7/2_ state, which may be involved in two-way energy transfer between the Sm^3+^ and MoO_6_ group, responsible for both Sm^3+^ emission and thermal quenching.

The shape of the emission spectra at a low temperature (77 K) differs significantly from the room-temperature emission spectra (Figure 4d). The most dominant lines at 77 K cannot be assigned directly to the ^4^G_5/2_ → ^6^H_J_ transitions, but are noticeably distant from the room-temperature emission lines by the same wavenumbers, which correspond to the phonon energies derived from the Raman spectrum (Figure 1e). This is uncommon for Sm^3+^ emission at low temperature, which usually features ^4^G_5/2_ → ^6^H_J_ transitions unaccompanied by their phonon-related components [35]. Apparently due to the high symmetry of the BMM structure, the ^4^G_5/2_ → ^6^H_J_ transitions are forbidden and must be coupled with phonons to occur. This causes the material to only exhibit anti-Stokes phonon-coupled transitions at low temperatures. When the temperature is higher, the lattice distortions allow for direct ^6^H_5/2_ → ^6^F_J_ transitions. This unique phenomenon, observed previously for only a handful of materials doped with transition metal ions, such as Cr^3+^ or Mn^4+^ [36,37], allow for very sensitive temperature sensing using the fluorescence intensity ration (FIR) between the phonon-coupled emission at 612 nm and regular ^4^G_5/2_ → ^6^H_9/2_ emission at 642 nm (Figure 4e). The relative sensitivity calculated [38] from this FIR has a maximum of 2.7% K^−1^ at −30 °C and another local maximum of 1.6% K^−1^ at 75 °C. Such a value is, to the best of our knowledge, one of the highest achieved for luminescent thermometry performed using only Sm^3+^ ions—comparable only to 3.83% K^−1^ obtained in Cs_4_PbBr_6_ borogermanate glass [39]—and significantly higher than, e.g., 0.5% K^−1^ for YAG [40] and 0.26% K^−1^ for Ba_2_Mg(PO_4_)_2_ [41]. Higher sensitivities were obtained only for materials exhibiting Sm-independent host emission, e.g., 7.08% K^−1^ in CaWO_4_ [42] or 10.14% K^−1^ for TiO_2_ nanoparticles [43]. The obtained result, resulting purely from the high symmetry of the BMM host, sets a promising precedent for temperature sensing using high-symmetry perovskite materials.

The emission spectra with dominant red emission results in CIE color coordinates located at (0.631, 0.370) (Figure 5a). Such color coordinates located towards the red part of the color coordinate scale, work efficiently in lowering the CCT and enhancing the CRI of the commonly used cool white LEDs. To demonstrate the impact of BMM:Sm emission on the luminescence of the wLED, its spectrum was added to the spectrum of the commonly used wLED with CRI equal 85 and CTT equal 4059 K. The result is a white emission with CRI of 91 and CCT of 2943 K constituting the warm white emission (Figure 5b).

## 3. Conclusions

Molybdates with a double-perovskite structure with the formula Ba_2_MgMoO_6_ doped with Sm^3+^ ions were successfully obtained for the first time using the co-precipitation method. The electric dipole (ED) ^4^G_5/2_ → ^6^H_9/2_ transition dominates; although, the Sm^3+^ ions are located in high symmetry, and the spectroscopic properties are characteristic for Sm^3+^ at octahedral sites with 6-fold coordination. Two-site Sm^3+^ emission was identified in the emission spectra of the 0.1% Sm^3+^ sample by comparing the relative intensity of the maxima at 597 nm and 642.6 nm. The obtained results suggest that in this sample Sm^3+^ ions are located in both Mg^2+^ and Ba^2+^ sites. At higher concentrations, the Ba^2+^ site is less preferable for Sm^3+^ doping, and as a result, the emission becomes more uniform and single-site. The aim of our work was to improve the properties of white-light-emitting diode lighting by supplying the missing red component and optimizing its correlated color temperature and color-rendering index. We can successfully conclude that the obtained BMM:Sm^3+^ materials are good candidates as a red phosphor, achieving a simulated CRI value of 91 and a CCT of 2943 K, giving a warm white emission. Temperature measurements brought us completely unexpected results. The relative sensitivity calculated from FIR is a maximum of 2.7% K^−1^ at −30 °C and another local maximum of 1.6% K^−1^ at 75 °C. This value is one of the highest achieved for luminescence thermometry performed only using Sm^3+^ ions. This is an extremely promising precedent for temperature sensing using perovskite materials with high symmetry that results from the high symmetry of the BMM host.

## Figures and Tables

**Figure 1 materials-17-01897-f001:**
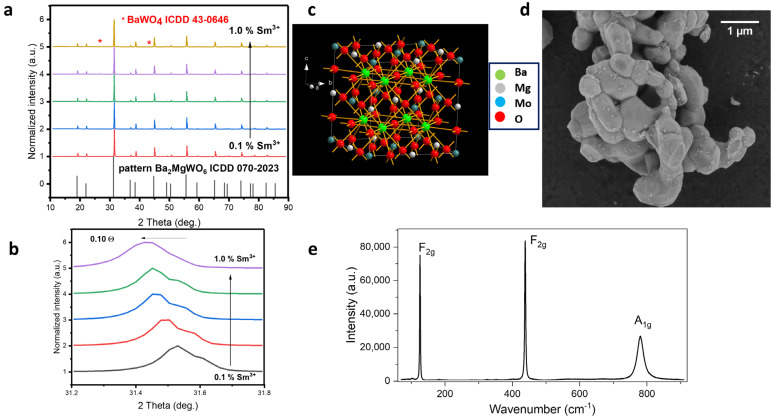
(**a**) XRD results of the studied dopuble-perovskite BMM doped with Sm^3+^, (**b**) XRD lines shift. (**c**) Unit cell of BMM with the space group *Fm*$\bar{3}$*m* (**d**) SEM image and (**e**) Raman spectra of the BMM: 1.0% Sm^3+^.

**Figure 2 materials-17-01897-f002:**
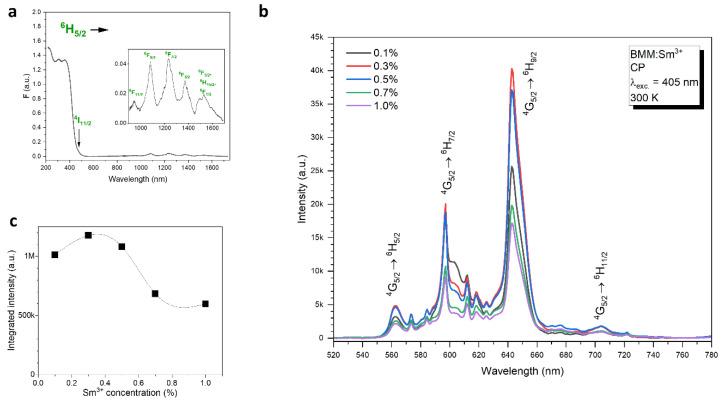
(**a**) Absorption spectra of the BMM: 1.0% Sm^3+^, (**b**) emission spectra, (**c**) integral emission intensity. The line in (**c**) serves only to guide the eye.

**Figure 3 materials-17-01897-f003:**
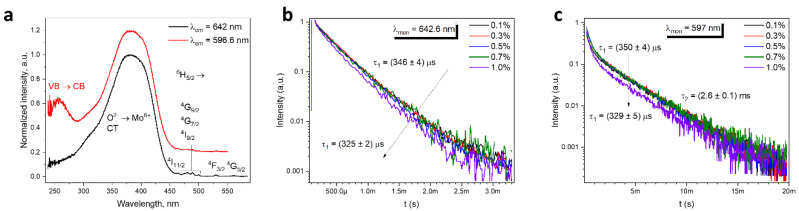
(**a**) Excitation spectra of the BMM:0.3% Sm^3+^ and (**b**,**c**) decay times of the BMM doped with Sm^3+^ monitored at 642.6 nm (**b**) and 597 nm (**c**).

**Figure 4 materials-17-01897-f004:**
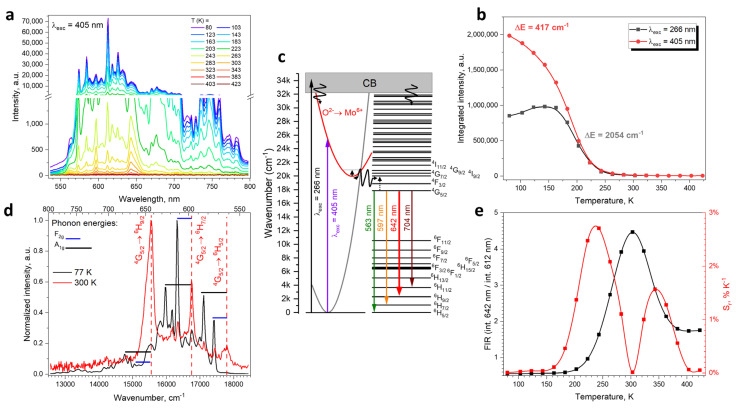
(**a**) Temperature-dependent emission spectra of BMM:0.3% Sm^3+^, (**b**) thermal quenching curves, and corresponding activation energies of BMM:Sm^3+^ under 405 and 266 nm excitation, (**c**) energy level scheme of the luminescent centers in Sm^3+^-doped BMM, (**d**) 77 K and 300 K emission spectra of BMM:Sm^3+^, (**e**) FIR and relative sensitivity of temperature sensing.

**Figure 5 materials-17-01897-f005:**
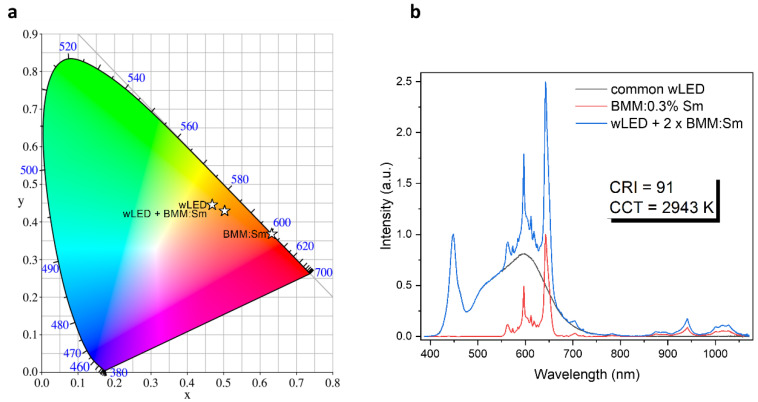
(**a**) CIE diagram of the BMM:0.3% Sm^3+^, (**b**) normalized emission spectra of common wLED, BMM doped with Sm^3+,^ and simulated luminescence of combined wLED and BMM:Sm^3+^.

**Table 1 materials-17-01897-t001:** Barycenters of energy levels of Sm^3+^ in BMM host determined from the excitation (^4^I_11/2_, …, ^4^G_5/2_), absorption (^6^F_11/2_, …, ^6^F_1/2_), and emission spectra (^6^H_13/2_, …, ^6^H_7/2_).

Level	Wavenumber
	(cm^−1^)
^4^I_11/2_	21,575
^4^G_9/2_	20,812
^4^G_7/2_	20,429
^4^I_9/2_	20,008
^4^F_3/2_	18,879
^4^G_5/2_	17,857
^6^F_11/2_	10,520
^6^F_9/2_	9163
^6^F_7/2_	7999
^6^F_5/2_	7129
^6^F_3/2_	6634
^6^H_15/2_	6492
^6^F_1/2_	6380
^6^H_13/2_	5076
^6^H_11/2_	3663
^6^H_9/2_	2302
^6^H_7/2_	1095
^6^H_5/2_	0

## Data Availability

No new data were created or analyzed in this study. Data sharing is not applicable to this article.

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
