# Peer review of "Highly Sensitive Temperature Sensors Resulting from the Luminescent Behavior of Sm3+-Doped Ba2MgMoO6 High-Symmetry Double-Perovskite Molybdate Phosphors"

_materials, 2024, doi:10.3390/ma17081897_

Round 1
Reviewer 1 Report
Comments and Suggestions for Authors
This manuscript reported that the luminescent properties of Sm3+-doped Ba2MgMoO6 for Highly Sensitive Temperature Sensors and wLEDs. The experimental results are interesting. However, there are still some issues that need to be clarified and corrected by the authors. The manuscript can be recommended for publication after a major revision. The issues are as follows:
1、In the fifth paragraph introduction part, the authors should provide the current research status of molybdate phosphors for temperature sensors, such as:
[1] Materials Today Chemistry, 2023, 32:101661
[2] Journal of Alloys and Compounds, 2022, 917: 165435.
[3] Journal of Alloys and Compounds, 2018, 738: 473-483.
2、Please modify the crystal structure diagram in Figure 1c. Removing excess ionic bonds and showing Mg and Mo atoms.
3、Without a standard crystal cif model, how did the author calculate the cell parameters?
4、“We determined the average 138 crystallite sizes in the ImageJ program and they are 0.62 µm.” Please draw a histogram of the average crystal size.
5、In Figure 4, please using K as the scales of temperature.
6、Please correct “Figure 3. a) excitation spectrum and b,c) decay times of the BMM doped with Sm3+ monitored at 191 642.6 nm (b) and 597 nm (c).”
Comments on the Quality of English LanguageMinor editing of English language required.
Author Response
Thank you very much for carefully checking my work " Highly sensitive temperature sensors resulting from the luminescent behavior of Sm3+-doped Ba2MgMoO6 high symmetry double-perovskite molybdate phosphors", N. Miniajluk-Gaweł, B. Bondzior, M. Ptak, P. J. Dereń, Manuscript ID: materials-2960975. All comments and opinions were extremely valuable to us and valuable for our work. Below are answers to questions from the editor and reviewers. all changes have been made to the Manuscript and Supporting Information. Changes have been made in red.
Comments and Suggestions for Authors (Reviewer 1)
This manuscript reported that the luminescent properties of Sm3+-doped Ba2MgMoO6 for Highly Sensitive Temperature Sensors and wLEDs. The experimental results are interesting. However, there are still some issues that need to be clarified and corrected by the authors. The manuscript can be recommended for publication after a major revision. The issues are as follows:
1、In the fifth paragraph introduction part, the authors should provide the current research status of molybdate phosphors for temperature sensors, such as: [1] Materials Today Chemistry, 2023, 32:101661, [2] Journal of Alloys and Compounds, 2022, 917: 165435, [3] Journal of Alloys and Compounds, 2018, 738: 473-483.
Answer: Thank you very much for your comment. Indeed, in the fifth paragraph introduction part, there is no information about the current state of research on molybdate phosphors for temperature sensors. Thank you for providing sample publications, but I did my own literature review. The following sentence was included in the work:
“The current state of research on molybdate phosphors for temperature sensors is not extensive and mainly concerns alkaline earth molybdates [17, 18, 19}. However, there is no information about double perovskites, so this work is very valuable and provides new information regarding non-contact thermometry.”
2、Please modify the crystal structure diagram in Figure 1c. Removing excess ionic bonds and showing Mg and Mo atoms.
Answer: Thank you very much for your comment. I included a corrected drawing of the BMM double perovskite unit cell in the work. I removed the bonds between atoms and presented Mg and Mo without octahedra. However, I think that showing the polyhedra is important for this structure because this material adopts a cubic structure with a rock salt lattice sharing the corners of the MgO6 and MoO6 octahedrons and with Ba cations placed in cubic 12-fold coordination sites.
3、Without a standard crystal cif model, how did the author calculate the cell parameters?
Answer: Thank you very much for your comment. The discussion of structural results includes information: “There is no standard pattern for the structure of BMM, but according to what was written in the previous publication [9], the structure of this material matches well with the ICDD 070-2023 pattern of Ba2MgWO6 (see Fig. 1a).”. Hence, the parameters of the Ba2MgMoO6 double perovskite unit cell were determined based on Ba2MgWO6 standard crystal cif model.
4、“We determined the average 138 crystallite sizes in the ImageJ program and they are 0.62 µm.” Please draw a histogram of the average crystal size.
Answer: Thank you very much for the comments. The supplementary contains a histogram of the average crystal size, determined on the basis of 100 crystallites (Fig. S1).
5、In Figure 4, please using K as the scales of temperature.
Answer: Thank you very much for the comments. The scales of the temperature were changed to K in Figures 4a and 4e.
6、Please correct “Figure 3. a) excitation spectrum and b,c) decay times of the BMM doped with Sm3+ monitored at 191 642.6 nm (b) and 597 nm (c).”
Answer: Thank you very much for the comments. The caption has been corrected to say “Fig. 3. a) Excitation spectra and b,c) decay times of the BMM doped with Sm3+ monitored at 642.6 nm (b) and 597 nm (c).”

Reviewer 2 Report
Comments and Suggestions for Authors
This article discusses the properties of Sm3+-doped double-perovskites Ba2MgMoO6. The morphology and luminescent properties of the synthesized materials have been investigated. I would like to mention the significant practical importance of the work, because it involves combining the synthesized material with a wLED. The article is well-written and should be accepted after minor revision by addressing the following comments:
1. At the graph in the Fig. 2c, the maximum is not clearly defined. I suggest to measure one more concentration such as 0.2 and 0.4 if it is possible.
2. I suggest to check the samarium content in the synthesized samples using EDX, AAS or AES. It would improve the quality of the work.
Author Response
Thank you very much for carefully checking my work " Highly sensitive temperature sensors resulting from the luminescent behavior of Sm3+-doped Ba2MgMoO6 high symmetry double-perovskite molybdate phosphors", N. Miniajluk-Gaweł, B. Bondzior, M. Ptak, P. J. Dereń, Manuscript ID: materials-2960975. All comments and opinions were extremely valuable to us and valuable for our work. Below are answers to questions from the editor and reviewers. all changes have been made to the Manuscript and Supporting Information. Changes have been made in red.
Comments and Suggestions for Authors (Reviewer 2)
This article discusses the properties of Sm3+-doped double-perovskites Ba2MgMoO6. The morphology and luminescent properties of the synthesized materials have been investigated. I would like to mention the significant practical importance of the work, because it involves combining the synthesized material with a wLED. The article is well-written and should be accepted after minor revision by addressing the following comments:
- At the graph in the Fig. 2c, the maximum is not clearly defined. I suggest to measure one more concentration such as 0.2 and 0.4 if it is possible.
Answer: Thank you very much for the comments. We agree, that the resolution of the optimal concentration determination is limited. However, from the intensity dependence on the concentration it is clear, that the maximum is between for concentration between 0.3 and 0.5% Sm3+. Also, the difference between the intensity values of those two, in our opinion, is small enough not to require to further narrow down the optimal concentration up to 0.1 resolution.
- I suggest to check the samarium content in the synthesized samples using EDX, AAS or AES. It would improve the quality of the work.
Answer: Thank you very much for the comments. The content of samarium and other elements included in the matrix was checked using EDX measurement. Theoretical and experimental compositions were compared and the results are presented in the table below. The lowest concentrations were not measured due to the lack of sensitivity of the equipment. According to the obtained results, it can be seen that the samarium content is slightly higher than expected. The results were not included in the paper.
Composition |
BMM:0.5% Sm3+ |
BMM:0.7% Sm3+ |
BMM:1.0% Sm3+ |
|||
TH [%] |
EX [%] |
TH [%] |
EX [%] |
TH [%] |
EX [%] |
|
Ba2+ |
55.92 |
59.99 |
55.92 |
48.89 |
55.92 |
58.42 |
Mg2+ |
4.93(52) |
6.40 |
4.92(53) |
18.17 |
4.91(04) |
7.50 |
Mo6+ |
19.54 |
18.79 |
19.54 |
14.99 |
19.54 |
18.53 |
O2- |
19.56 |
14.11 |
19.56 |
17.05 |
19.56 |
14.36 |
Sm3+ |
0.15(31) |
0.71 |
0.21(43) |
0.9 |
0.30(62) |
1.19 |
TH- theoretical composition; EX- experimental composition

Reviewer 3 Report
Comments and Suggestions for Authors
This paper reports the spectroscopic properties of Sm3+ doped Ba2MgMoO6 for temperature sensing and WLED application as red component.
These phosphors were mainly characterized by XRD, SEM and Raman. Optical properties were reported and discussed considering the proposed applications.
This article reports relevant features and can be accepted for publication if the following issues and questions are considered:
- Some PLQY measurements should be provided and compared with other red phosphors.
- Lines 193-201: It is stated that the longer lifetime values (Fig. 3c) correspond to Sm3+ ions located in the Ba site. However this site is not supposed to be occupied for higher Sm amounts as suggested in lines 158-177. How is it possible to record decay curves for the 597 nm emission for higher Sm3+ contents?
- Fig. S1 is missing (line 205). Can activation energies be given with a cm-1 unit?
- Fig. 4a: several emission components located near 750 nm are observed at low temperature in Fig. 4a. What is their origin?
- Line 234: The 4.5% K-1 relative sensitivity is wrong due to a confusion with the FIR curve. The following discussion and comparison with other phosphors must be modified. See also conclusion and abstract.
Herefater some minor required corrections:
- Introduction lines 48-49: RE dopants are certainly not located in MoO42- or MoO66- sites.
- Section 2.1 line 87: the Sm3+ higher amount is 1%, not 10% (according to Fig. 1).
- line 96: it is not possible to evaporate a precipitate. The authors certainly wanted to talk about water evaporation.
- Section 3.1 lines 129-130: Considering a 6-fold coordination for Mg2+ ions, the correct IR for Mg2+ and Sm3+ are 72 and 96 pm respectively (see Shannon and Prewitt article in Acta Cryst.).
- Fig. 1: a brown colour is needed for Mg, not grey (colour code). Please give a b c axes in Fig. 1c.
- What doped sample is shown in Fig. 1d? Same question for Fig. 1e, Fig. 2a, Fig. 3a, Fig. 4a, Fig. 5a.
- Fig. 2a: the 6H5/2 - 4I11/2 absorption line is invisible and can be removed from this Figure.
- Line 172: Ba2ZnWO6 not BaZn2WO6.
- Line 200: SrAl4O7 not SrAl2O4. Moreover Sr2+ ions are are located in a 7-fold coordination in SrAl4O7 and 7- and 8-fold coordinations in Sr2SiO4. Do you mean that lifetime increases when coordination increases? Please explain this assertion.
- Give nm scale in Fig. 4d.
- Ref 10: pages are missing.
Author Response
Thank you very much for carefully checking my work " Highly sensitive temperature sensors resulting from the luminescent behavior of Sm3+-doped Ba2MgMoO6 high symmetry double-perovskite molybdate phosphors", N. Miniajluk-Gaweł, B. Bondzior, M. Ptak, P. J. Dereń, Manuscript ID: materials-2960975. All comments and opinions were extremely valuable to us and valuable for our work. Below are answers to questions from the editor and reviewers. all changes have been made to the Manuscript and Supporting Information. Changes have been made in red.
Comments and Suggestions for Authors (Reviewer 3)
This paper reports the spectroscopic properties of Sm3+ doped Ba2MgMoO6 for temperature sensing and WLED application as red component.
These phosphors were mainly characterized by XRD, SEM and Raman. Optical properties were reported and discussed considering the proposed applications.
This article reports relevant features and can be accepted for publication if the following issues and questions are considered:
- Some PLQY measurements should be provided and compared with other red phosphors.
Answer: Thank you very much for the comments. For reasons currently beyond our control, the PLQY of this sample could not be recorded using equipment available to us.
- Lines 193-201: It is stated that the longer lifetime values (Fig. 3c) correspond to Sm3+ions located in the Ba site. However this site is not supposed to be occupied for higher Sm amounts as suggested in lines 158-177. How is it possible to record decay curves for the 597 nm emission for higher Sm3+contents?
Answer: Thank you very much for the comments. The statement in lines 158-177 was phrased incorrectly and misleading. It suggested that it was impossible for Sm3+ ions to enter Ba sites at high concentration. The sentence has been changed to say: “For higher concentrations, as the lattice constant increases the Ba2+ site becomes less preferable for Sm3+ dopant and as a result the emission becomes more uniform and seemingly one-site.” The relevant changes were also made in Abstract and Conclusions.
- Fig. S1 is missing (line 205). Can activation energies be given with a cm-1 unit?
Answer: Thank you very much for the comments. Figure S1. Fitting of an Arrhenius equation to temperature-dependent emission intensity of BMM:Sm3+ excited at 405 nm (left) and 266 nm (right) was presented in supplementary. However, to answer the second question, the activation energy can be presented in any energy units, but it is usually given in cm-1 or eV.
- Fig. 4a: several emission components located near 750 nm are observed at low temperature in Fig. 4a. What is their origin?
Answer: Thank you very much for the comments. The energy of this transition suggests, that these lines are most likely resulting from transition 4G7/2→6F5/2. It is observed only at low temperature, when the relaxation to 4G5/2 emitting state is hindered.
- Line 234: The 4.5% K-1 relative sensitivity is wrong due to a confusion with the FIR curve. The following discussion and comparison with other phosphors must be modified. See also conclusion and abstract.
Answer: Thank you very much for the comments. We appreciate this important correction. The value has been corrected throughout the manuscript. The comparison with other phosphors was modified to say “one of the highest” instead of “the highest”, as it is not in fact the highest achieved in Sm-doped material.
Herefater some minor required corrections:
- Introduction lines 48-49: RE dopants are certainly not located in MoO42-or MoO66-
Answer: Thank you very much for the comments. Of course the reviewer is right that, RE dopants are certainly not located in MoO42- or MoO66- sites. I'm very sorry for the error. The publication included the correct sentence: “It is apparent that the rare earth dopants tend to locate themselves in highly symmetrical octahedral MgO6 sites, which is credited for their unique luminescence properties in form of almost monochromatic emission spectra [8; 9].”
- Section 2.1 line 87: the Sm3+higher amount is 1%, not 10% (according to Fig. 1).
Answer: Thank you very much for the comments. I agree that the highest concentration of Sm3+ ion is 1%, not 10%. The mistake was corrected in publication.
- line 96: it is not possible to evaporate a precipitate. The authors certainly wanted to talk about water evaporation.
Answer: Thank you very much for the comments. I am very sorry for the language error. Of course, the precipitate cannot be evaporated, only the water from it. The corrected sentence was included in the publication:
“The water was evaporated and the precipitate was dried by heating at 80°C for 20 hours and then pre-sintering at 600°C for 12 hours.”
- Section 3.1 lines 129-130: Considering a 6-fold coordination for Mg2+ions, the correct IR for Mg2+ and Sm3+ are 72 and 96 pm respectively (see Shannon and Prewitt article in Acta Cryst.).
Answer: Thank you very much for the comments. I agree with the reviewer's opinion that according to Shannon's article (Acta Cryst. (1976).A32, 751), the ionic radius of 6-fold coordination Mg2+ and Sm3+ ions are 72 and 96 pm, respectively. However, in my work I provided the values (Mg2+, IR = 86 pm and Sm3+, IR = 109 pm). according to the website “web of elements”, which is also a reliable source of information on ionic radii.
- 1: a brown colour is needed for Mg, not grey (colour code). Please give a b c axes in Fig. 1c.
Answer: Thank you very much for the comments. The corrections have been applied to the drawing.
- What doped sample is shown in Fig. 1d? Same question for Fig. 1e, Fig. 2a, Fig. 3a, Fig. 4a, Fig. 5a.
Answer: Thank you very much for the comments. With the exception of Fig 1a (0.1%) and 2a (1.0%), all the listed figures were of the BMM:0.3% Sm3+. The figure captions were corrected accordingly.
- Fig. 2a: the 6H5/2- 4I11/2absorption line is invisible and can be removed from this Figure.
Answer: Thank you very much for the comments. As per suggestion, the 6H5/2 - 4I11/2 absorption line was removed from Fig. 2a.
- Line 172: Ba2ZnWO6not BaZn2WO6.
Answer: Thank you very much for the comments. Of course, the formula was written correctly.
- Line 200: SrAl4O7not SrAl2O4. Moreover Sr2+ions are are located in a 7-fold coordination in SrAl4O7 and 7- and 8-fold coordinations in Sr2SiO4. Do you mean that lifetime increases when coordination increases? Please explain this assertion.
Answer: Thank you very much for the comments. The incorrect material formula has been corrected. The provided examples were meant to indicate the difference and the tendency of Sm in octahedral sites to exhibit shorter emission decay than those in non-octahedral sites. The sentence was rewritten for clarity: “(…) observed commonly for Sm3+ - doped materials with site coordination number other than 6, such as SrAl4O7 [33] and Sr2SiO4 [34].”
- Give nm scale in Fig. 4d.
Answer: Thank you very much for the comments. The nm scale was given in Fig. 4d as secondary scale at the top. The primary scale in cm-1 is necessary to illustrate the energy separation of the vibronic components.
- Ref 10: pages are missing.
Answer: Thank you very much for the comments. I apologize for the oversight. Page numbers have been entered.

Round 2
Reviewer 3 Report
Comments and Suggestions for Authors
This paper can be accepted for publication.